# Structural effects of high laser power densities on an early bacteriorhodopsin photocycle intermediate

Quentin Bertrand [1], Przemyslaw Nogly [1,2,9], Eriko Nango [3,4,5], Demet Kekilli[1], Georgii Khusainov[1], Antonia Furrer [1], Daniel James[1], Florian Dworkowski [6], Petr Skopintsev[1], Sandra Mous[1,2], Isabelle Martiel[6], Per Börjesson [7], Giorgia Ortolani [7], Chia-Ying Huang [6], Michal Kepa [1], Dmitry Ozerov [6], Steffen Brünle[1], Valerie Panneels [1], Tomoyuki Tanaka[3], Rie Tanaka[3], Kensuke Tono [4], Shigeki Owada[3], Philip J. M. Johnson [6], Karol Nass[6], Gregor Knopp [6], Claudio Cirelli [6], Christopher Milne [6], Gebhard Schertler [1], So Iwata [5,8], Richard Neutze [7], Tobias Weinert [1] ✉ & Jörg Standfuss [1] ✉

Time-resolved serial crystallography at X-ray Free Electron Lasers offers the opportunity to observe ultrafast photochemical reactions at the atomic level. The technique has yielded exciting molecular insights into various biological processes including light sensing and photochemical energy conversion. However, to achieve sufficient levels of activation within an optically dense crystal, high laser power densities are often used, which has led to an ongoing debate to which extent photodamage may compromise interpretation of the results. Here we compare time-resolved serial crystallographic data of the bacteriorhodopsin K-intermediate collected at laser power densities ranging from 0.04 to 2493 $GW/cm^2$ and follow energy dissipation of the absorbed photons logarithmically from picoseconds to milliseconds. Although the effects of high laser power densities on the overall structure are small, in the upper excitation range we observe significant changes in retinal conformation and increased heating of the functionally critical counterion cluster. We compare light-activation within crystals to that in solution and discuss the impact of the observed changes on bacteriorhodopsin biology.

Time-resolved serial crystallography (TR-SFX) at X-ray free electron lasers enables the study of protein dynamics at atomic spatial and femtosecond temporal resolution, potentially capturing conformational intermediates directly[1,2]. While it is possible to trigger reactions using rapid mixing in the millisecond range[3,4], the technique has been particularly successful in studying light-activated proteins undergoing fast conformational changes when exposed to light. In this way, TR-SFX provided valuable insights into the molecular mechanisms of various biological processes, including DNA repair[5,6], photosynthesis[7–10], ion transport[11–16], and vision[17].

An experimental challenge of TR-SFX is the high optical density of crystals resulting from tight packing and typically high extinction coefficients of light-activatable proteins. Even when using intense lasers focused on micrometer-sized crystals, the reaction is still triggered heterogeneously in most cases. The laser fluences (i.e., energy per unit area) used during an experiment are crucial as they define the number of photons delivered per chromophore and determine the proportion of activated molecules that can be obtained. However, the delivery of too many photons in a short time span increases the

probability of the absorption of a second or third photon by the already excited state, potentially causing higher state excitation, photodamage, and reaction pathways that differ from the natural photochemical reaction. This is particularly true when the high laser power densities (i.e., laser energy per area and time) of femtosecond lasers are used to study the ultrafast temporal domain.

Depending on the laser power density, crystal size, and the absorption characteristics of the excited state, these multiphoton effects may divide the population of triggered reactions in a crystal into different species. It is important to mention that such alterations may also happen when the laser pulse length exceeds the lifetime of the excited state, and additional photons are absorbed by ground-state photo-intermediates. In some cases, this can have physiological relevance, in channelrhodopsins, for example, multiple excitations are needed to achieve full -activation, whereas continuous illumination leads to alternative desensitized photocycles[18].

One can think of the crystal as having several layers, each receiving a different number of photons and producing a separate diffraction pattern. Distant crystal layers contain different proportions of triggered protein, but each layer yields a diffraction pattern that is perfectly superposed on the other and hence contributes to observed diffraction intensities and, ultimately, the determined electron densities used for interpreting the results. In TR-SFX data analysis, this is usually neglected, making the simplified assumption that the diffraction signal arises from a mixed population of states that are equally distributed throughout the crystal. The average number of photons delivered as a function of penetration depths within the crystal can be calculated. However, comparing different experimental setups based on calculations alone can be challenging, as the measured parameters are often not standardized or vary throughout the experiment[19]. Therefore, analyzing and comparing TR-SFX data of the same target protein obtained under varying conditions may shed light on the range of laser power densities that can be used to reach the best balance between sufficient activation and photodamage.

Bacteriorhodopsin (bR) is an ideal target for studying laser-induced damage, as it is among the first proteins that have been abundantly studied by both freeze-trapping of structural intermediates and TR-SFX experiments in the femto-to-millisecond range[20,21]. Structurally, bR shares the seven trans-membrane α-helical topologies typical for other rhodopsins, as well as G protein-coupled receptors. The primary function of bR is that of a light-driven proton pump, where *trans* to *cis* isomerization of a covalently bound retinal chromophore starts a series of conformational changes necessary for directional transport against a concentration gradient. The K-intermediate is of particular interest for the study of photodamage as in this early photocycle intermediate, the retinal molecule has just crossed the conical intersection into the ground state but still retains some of its energy in a strained non-planar *cis* conformation[11,21,22]. The temporal range where the K-intermediate forms are furthermore accessible using both nano- and femtosecond lasers. Together with the wealth of information available for this prototypical membrane pump[23] and the spectroscopic characterization of high excitation regimes[24], this makes the K-intermediate an excellent candidate to investigate experimentally the structural impact of excessive laser excitation.

Spectroscopic studies have demonstrated that the isomerization behavior of the bR-bound retinal molecule is influenced by high excitation densities in two regimes. The initial range is marked by a gradual increase in the $S_1$ excited state decay time that is attributed to the internal conversion of higher excited states back into $S_1$[25]. However, at high power densities, the spectrum is redshifted, suggesting the formation of a different photoproduct present on picosecond to nanosecond timescales[24,26]. The effect of increasing fluence on the bR photocycle has been confirmed by time-resolved spectroscopy on bR crystals in recent experiments using nanosecond laser excitation[27]. The

structural effect of high laser power density on time-resolved serial crystallographic structures remains largely unknown. However, scientific discourse suggests that the impact of high laser power densities on serial crystallographic data needs to be thoroughly investigated[1,28,29] to understand the degree to which the absorption of multiple photons by an already excited chromophore influences observed structural intermediates and their interpretation. While it is sometimes possible to observe fluctuations in atomic positions in femtosecond crystallographic data and potentially draw conclusions on vibrational frequencies[16,29–31], these oscillations may be influenced by multiphoton effects. Indeed, a recent re-investigation of ultrafast changes upon carbon monoxide release from myoglobin[32] demonstrates subtle effects in the extent (~ 0.2 Å) and kinetics (~ 200 fs) of the reaction once laser power densities leave the linear regime where multi-photon effects can be excluded. Since the myoglobin-CO system has not evolved to be photoactive, it is an important question if such effects would propagate into later intermediates of a biologically relevant reaction.

Here, we investigate the effects of increasing laser power density on the light-driven proton pump bacteriorhodopsin with a focus on the K-intermediate that accumulates just after the excited retinal chromophore returns to the electronic ground state. Based on data collected with both femto- and nanosecond lasers, we discuss the energy dissipation inside crystals as well as the impact of laser power densities ranging from 0.04 to 2493 GW/cm² on retinal isomerization and a cluster of water molecules involved in proton transport. The observed structural effects provide an example of how and when multiphoton absorption in the femtosecond domain can influence later structural intermediates and their biological interpretation.

## Results
### Global effects on the bR structure

The following analysis is made based on several datasets collected at different X-ray Free Electron Laser sources under a broad range of excitation conditions (Supplementary Table 1). In addition to data collected at the Spring-8 Angstrom Compact Free Electron Laser (SACLA) and the Swiss X-ray Free Electron Laser (SwissFEL), we include previously published datasets from the Linac Coherent Light Source (LCLS)[16] and SACLA[11] that we re-analyzed for consistency (see Supplementary Fig. 1 for a comparison of data quality and effect of data collection at different facilities). While another structure of the K-intermediate (using a power density of 638 GW/cm² at LCLS) has been reported[29], we have excluded this data since the low number of collected images led to weak difference signals (Supplementary Fig. 2) and prevented conventional crystallographic refinement.

A number of simplifications are required to compare these data because the number of photons reaching each chromophore depends on several factors, including the direction of illumination, shadowing, scattering from the extruded cubic phase, or the size and orientation of crystals (Fig. 1A). These effects complicate both the direct comparison to spectroscopic data and between different TR-SFX experiments. For consistency, we, therefore, focus on laser power densities that we calculated from well-known experimental parameters (given in Supplementary Table 2) without taking light scattering effects into account that have been estimated in a range from 20% to 99%[16,17,19,29,33,34] depending on sample delivery method and the optical properties of the sample (e.g., if opaque or not) after preparation for injection. For LCP, the medium used for data collection of each experiment we analyzed, scattering contributions of 20%[29] and 35%[17] have been experimentally determined. We further rely on pairwise comparisons, for example, between the highest and lowest laser power settings used at SwissFEL, or by comparing femto- and nanosecond laser excitation under the same fluence at SACLA to reduce the impact of other experimental factors from measurements at different facilities. The overall structures are very similar in the temporal range from

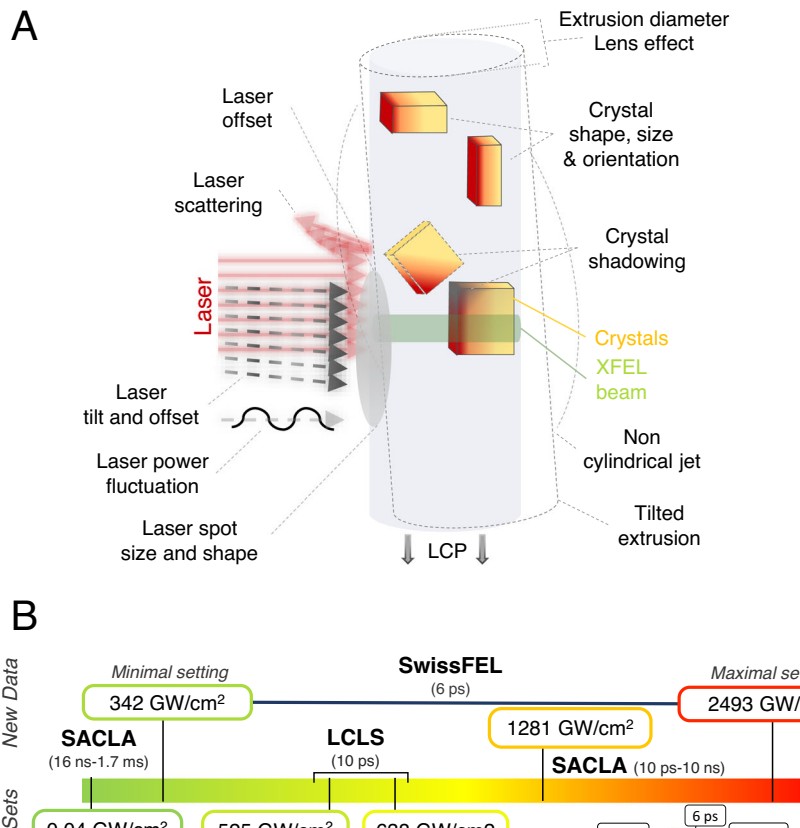

**Fig. 1 | TR-SFX data collection and dataset summary. A** Schematic representation of effects that could alter the laser power density transmitted to crystals in a TR-SFX experiment. The average path length in the crystals used was 7 µm. **B** Laser power range and X-ray Free Electron Lasers were used to derive the K-intermediate structures we compare in our study. All laser power densities have been calculated based on direct experimental parameters without taking uncertain factors into account. All datasets discussed in our study were recorded at a time delay where bR is expected to be in the K-intermediate. The next intermediate, L, appears later in the photocycle and becomes visible in datasets recorded at 290 ns[11].

6 ps to 16 ns where the K-intermediate of bR is dominant (Fig. 1B), as indicated by their low root mean square deviations (rmsd) (Supplementary Table 3A).

In addition to studying the effects of increasing laser power density, we combined data collected after femtosecond and nanosecond laser excitation to a logarithmic time series ranging from 10 ps to 1.7 ms. This series, collected at the same intermediate laser fluence, allows us to track global laser effects by comparing relative atomic B-factors and changes in unit cell volume over time. This provides insights about the experimental impact of delivering the same photon dose with either a short (femtosecond) or long (nanosecond) laser pulse. B-factor increase in the light data, when compared to their respective dark datasets, are moving from the region around retinal towards the surface of the protein (Fig. 2A) in the pico- to early nanoseconds. This is in line with heat diffusivity observed spectroscopically in proteins[35]. Heat distribution within the protein is followed by a rapid expansion of the unit cell, reaching its peak after 40 ns before fully relaxing back in microseconds (Fig. 2B). This later effect is due to the dissipation of heat throughout the crystal as the observed expansion compares well to those observed in temperature-jump crystallography[36] and time-resolved X-ray solution scattering[37].

### Effects on retinal isomerization

The primary reaction from the bR dark state to the K-intermediate is the isomerization of retinal from the all-*trans* conformation to a twisted 13-*cis* conformation that stores a significant part of the initial photoenergy[21,22]. The corresponding structural changes are visible in all of our difference density maps, but we observe additional effects from increasing laser power densities (Fig. 3, see Supplementary Fig. 1 for different contour levels). Reference difference density maps were recorded at 0.04 GW/cm² with a nanosecond laser (Supplementary Fig. 2), for which direct multiphoton effects on the excited state are highly improbable, although the 13-*cis* retinal ground state may be photoexcited and return to an all-trans configuration. From these data, we observe the characteristic negative to positive density pairs corresponding to the move of retinal C20, as anticipated for the K-intermediate[21]. The data recorded at 342 GW/cm² using a femtosecond laser exhibits the same retinal movement with comparable difference densities, albeit the density is more defined, likely because of the higher spatial and temporal resolution as well as a higher multiplicity of the data. We observe that the positive difference density associated with the canonical K-intermediate retinal C20 movement decreases above 525 GW/cm², suggesting the emergence of heterogeneity in the C20 movement between the 525 to 1281 GW/cm² range which already decreases the clarity of the difference signal. Despite the decrease in signal quality at 1281 GW/cm², the induced retinal conformation remains similar from 10 ps to 10 ns and compares well to the retinal conformation of the nanosecond laser control (Supplementary Fig. 3 and Supplementary Table 4). At 2493 GW/cm², this heterogeneity increases further as the signal associated with the C20 movement in

the K-intermediate continues to diminish. Interestingly, a new positive difference density peak appears above the dark retinal, potentially indicating the displacement of the retinal C20 toward this position.

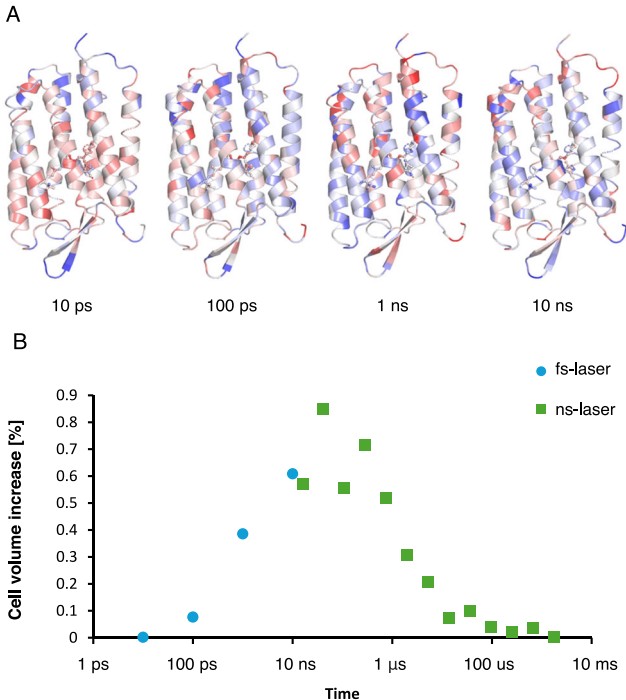

**A**

10 ps    100 ps    1 ns    10 ns

**B**

**Fig. 2 | Heat deposition into the protein and its dissipation into crystals.** **A** Changes in atomic B-factor between laser triggered and dark data (blue = decrease, red = increase) plotted on the structure of bR (cartoon). **B** Changes in unit cell volume over time indicate heat dissipation within crystals. Measurements using a femtosecond laser (blue) and nanosecond laser (green) were done at the same fluences.

To further elucidate the retinal movements, we computed extrapolated data using activated fractions using the retinal methyl C20 position with the methodology described in Pandey et al.[38] (Fig. 3). Up to a laser power density of 1281 GW/cm² refinements against these data resulted in a single retinal conformation with similar rmsd values to their respective dark state (Supplementary Table 3B). After refinement against data collected at 2493 GW/cm², residual $F_{ext}$(light)-$F_{calc}$ difference density indicates the presence of another retinal conformation (Supplementary Fig. 4) in agreement with the distinct positive $F_{obs}$(light)-$F_{obs}$(dark) difference density peak emerging at high laser power densities (Fig. 3). This new conformation is more planar and similar to retinal in the L-intermediate structures from Nango et al.[11]. but the low occupancy of this species and its convolution with the K-intermediate do not allow us to discern the exact atomic coordinates of either the retinal or the protein associated with this photodamage artifact. Overall, it is clear that the lower range of laser power densities that we examined in our study results in a consistent K-intermediate and does not display the effects on electron density signal and retinal conformation observed under the highest laser power density.

## Effects on counterion network

Besides retinal isomerization as the first step after photon absorption, the function of bR as a proton pump also requires early changes in the associated counterion network[39,40]. The network is formed by Asp85, Asp212, and Arg82, water molecules 400, 401, and 402 (Fig. 4), and is critical for stabilizing the proton on the retinal Schiff base in the dark state[41]. Disruptions to the structure or dynamics of the water cluster could thus potentially divert the photocycle along a non-physiological pathway.

In contrast to signals from the retinal that diminish at higher laser power densities, difference density maps around the counterion cluster show an increased magnitude of negative signal with increasing laser power (Fig. 4, see Supplementary Fig. 1 for different contour levels). Nevertheless, the positive differences around water 402 are also amplified, in agreement with the movement of water 402 in the

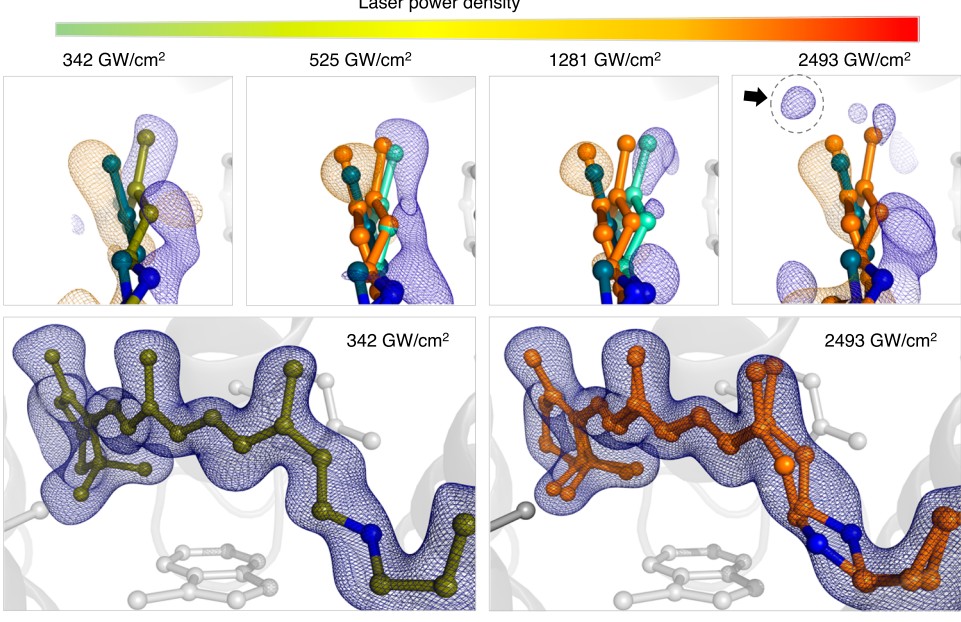

Laser power density

342 GW/cm²        525 GW/cm²        1281 GW/cm²        2493 GW/cm²

342 GW/cm²                        2493 GW/cm²

**Fig. 3 | Effect of increasing laser power densities on K-intermediate retinal conformation.** Difference electron density maps ($F_{obs}$(light)-$F_{obs}$(dark), negative density in yellow, positive density in blue, contoured at 3 σ) collected at different laser power densities. The lowest and highest laser power densities are a direct comparison using data collected at SwissFEL (6 ps). The two middle panels were collected at LCLS (10 ps) and SACLA (10 ps), respectively, and are shown for comparison. A distinct peak, not present in the 342 GW/cm² map, appears above the

negative density in the 2493 GW/cm² map (gray arrow), indicating a second retinal conformation. The models represented are: the refined dark model (dark green); the refined light model (light blue or light green for the model recorded at 342 GW/cm²) and the two structures obtained at the highest laser power density (orange) for comparison. The lower panel shows the refined retinal models and extrapolated density maps ($2F_{ext}$(light)-$F_{calc}$ at 1 σ) for the highest and lowest laser power density collected after femtosecond laser excitation.

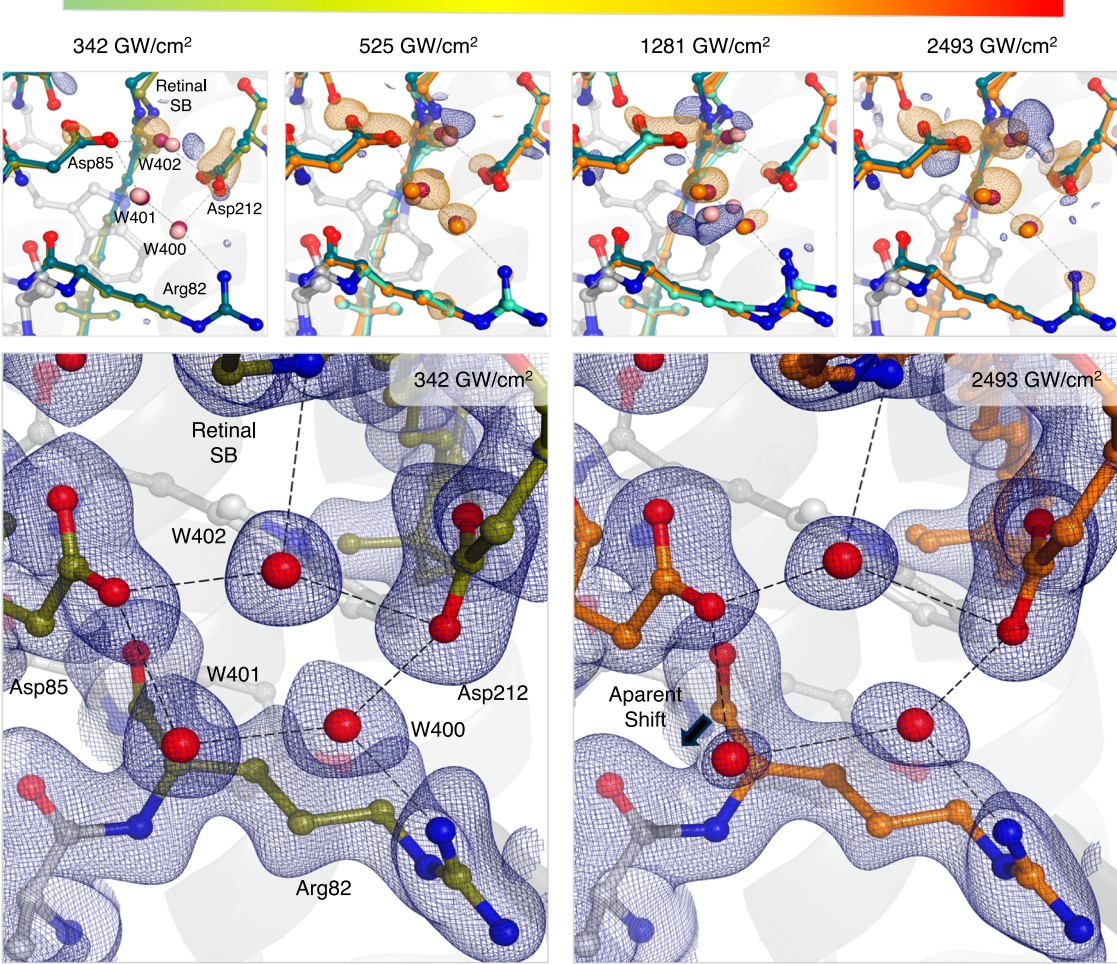

**Fig. 4 | Effect of increasing laser power densities on the K-intermediate counter ion cluster.** The lowest and highest laser power densities are a direct comparison using data collected at SwissFEL (6 ps). The two middle panels were collected at LCLS (10 ps) and SACLA (10 ps), respectively, and are shown for comparison. Difference electron density maps ($F_{obs}$(light)-$F_{obs}$(dark), negative density in yellow, positive density in blue, contoured at 3 σ) collected at different laser power densities. Each panel includes dark (dark green), light at low laser power (in light green), light (light blue) models, and the damaged structure obtained with the highest laser power density (orange). All models are aligned based on their corresponding dark models. Waters in the dark and light models are marked in purple and pink, respectively. The lower panel shows the refined models and extrapolated density maps ($2F_{ext}$(light)-$F_{calc}$ at 1 σ) for the highest and lowest laser power density collected after femtosecond laser excitation. Retinal SB = retinal Schiff base.

nanosecond laser-derived K-intermediate at the lowest laser power density (Supplementary Fig. 2).

Besides the shift in water 402, we observe an increase of negative density in water 401 and particularly around water 400, suggesting an increased mobility of these water molecules. At higher laser power densities, we further observe stronger negative difference densities around Asp85 and Arg82 and emerging positive difference densities. Since the Arg82 movement is typical for the M-intermediate; we investigated and were able to exclude light contamination due to increased scattering at higher laser fluences as a cause of these difference map signals (Supplementary Fig. 5). Instead, we attribute the changes to an anisotropic thermal motion.

The thermal load can be followed over time when analyzing the atomic B-factor difference between light and dark datasets collected at 10 ps, 100 ps, 1 ns, and 10 ns delays excited at 1281 GW/cm² (Supplementary Fig. 6A). The relative B-factor difference on the counterion cluster and retinal continually decays before disappearing after 10 ns (Supplementary Fig. 6B) where the heat is nearly evenly distributed throughout the crystal as evident from the maximally expanded unit cell (Fig. 2B). While a B-factor increase of about 4 percent suggests a moderate thermal load on the cluster, this load will not only be present in the isomerized, light-activated fraction of bR but also to the fraction that did not isomerize. Hence, in extrapolated densities where the activated fraction has been determined on the directed retinal motion, the corresponding electron densities on the counterion cluster are decreased and the positional accuracy of the refined atomic position decreases (Supplementary Fig. 7). As the heat is dissipated throughout the crystal, the waters of the counterion cluster begin to settle back to their original position from late picoseconds to early nanoseconds, a trend that is also reflected in the decreasing negative difference map signal on the water cluster from 10 ps to 10 ns (Supplementary Fig. 3).

Interestingly, we do not observe strong variations in the structural models obtained from these datasets, despite the additional negative difference peaks at higher laser power densities. Only at the highest laser power density, refinements suggest a displacement of water 401 by 0.7 Å (Fig. 4), which we attribute to a refinement artifact resulting from a decrease in extrapolated density and decreased positional accuracy described above. Nevertheless, given the critical roles of this cluster of water molecules and Asp85 in the proton transport pathway, there is a profound effect of high photon energies on this functionally

critical element, that reverses in the nanosecond range due to heat dissipation.

## Discussion

The question of how best to trigger a photoactive ligand in a TR-SFX experiment will remain important because the answer depends on many factors, including extinction coefficients of the dark and photoactivated states, quantum efficiency, and the temporal domain that is under investigation. Low laser fluences can be sufficient if maintained long enough. For example, we had good success in using millisecond illumination with laser diodes to probe late bR photointermediates[42] or to probe dynamics in the A2a receptor, even though the stilbene-based photochemical affinity switches had reduced quantum efficiency within the protein binding pocket[43]. Nanosecond lasers are another popular choice[44] because much of the biologically relevant temporal domain can be covered, significant absorption of additional photons by short-lived excited states can be mostly excluded, and lower quantum efficiencies can be balanced[45]. The investigation of ultrafast photoreactions such as retinal isomerization, on the other hand, necessitates the use of higher laser power densities because a similar number of photons needs to be compressed into a much shorter pulse.

In comparison to other ultrafast TR-SFX experiments, the highest laser power density in our series is only surpassed by a study focusing on the primary response of a phytochrome[33]. Spectroscopic studies on phytochrome show significant effects of high laser power densities[46], pointing out that the chemical nature of the ligand (e.g., ionization energy) must always be considered, and that different systems react differently to high laser power activation. Other studies employing higher power densities than presented here, typically shifted the laser to avoid activating a large area of crystals upstream of the X-ray interaction region, lowering the intensity delivered to the sample at the interaction zone[16,29]. The lower end of our scale is comparable to two other femtosecond laser experiments, one conducted at 135 GW/cm² on the photosynthetic reaction center[7] and one on carbon monoxide release from myoglobin where significant signals have been observed down to 105 GW/cm² due to the exceptionally high quantum efficiency of this system[32]. Since laser scattering is reported with various strengths and calculated differently, we excluded light scattering effects from the reported power densities. Instead, we relied solely on experimentally measured laser parameters for a more accurate comparison. In our study, all datasets were recorded with similar experimental setups, which allows us to consider the difference in laser scattering between experiments as minimal.

While differences in experimental setups or the photochemical parameters of the investigated reactions make it hard to generalize findings, there are some general conclusions to be drawn from the bR system where time-resolved measurements have been done using a wide range of lasers. It is generally an adapted practice to 'Stay away from ESA', i.e., minimize the overlap between excitation wavelength and excited state absorption (ESA). This simple rule of thumb has helped minimize direct multiphoton effects from using femtosecond lasers. However, at the highest laser power densities in our bR experiments, this was not sufficient to prevent the direct deposition of heat or its formation through internal conversion between $S_n$ and $S_1$ exited states (Fig. 5A, B). Once formed, the heat dissipates into the crystal on the picosecond time scale, in line with cooling rates observed in spectroscopic studies[35]. The result is a weakening of positive difference densities due to undirected vibrational motions and potential retinal distortions at the highest settings. The thermal energy responsible for increased heterogeneity starts to be dissipated from the crystals in nano- to microseconds, which is in line with the effects seen in T-jump studies[36]. The cell expansion and B-factor we observe suggest a heating effect in the order of 10–20 K when calibrated against temperature effects on lysozyme crystals[47], which is about an order of magnitude less than the theoretical heat load suggested based on the heat capacity of proteins and the number of incident photons[28]. Such laser-induced heating effects may especially affect early nanosecond time-resolved data due to cell expansion and potentially blur difference map signals, especially if the cell expansion is anisotropic or molecules shift inside the unit cell.

On the purely structural level, our results show that even at the highest laser power densities, the overall conformation of the K-intermediate remained basically unchanged compared to structures obtained using nanosecond laser excitation (Supplementary Table 3) and freeze-trapping experiments[21]. The regime where we start to observe increased heterogeneity in difference maps begins at 525 GW/cm² leading up to the emergence of a non-natural retinal structure at the highest settings (Fig. 3). The power densities at which we begin to observe structural changes further differ from those reported in spectroscopic laser power titrations on bR in solution[26], where an altered photoproduct is formed at power densities of 100 GW/cm², and larger absorption changes have been reported around 200 GW/cm². These spectroscopic changes may be due to the altered retinal conformation we observe at 2493 GW/cm², suggesting an order of magnitude difference between both measurements, which can be explained by the multiple effects, including scattering shown in Fig. 1A. UV/Vis, infrared, and Raman spectroscopy are furthermore highly sensitive techniques capable of detecting electronic and vibrational changes of excited chromophores in great detail, while time-resolved crystallography is less sensitive in this regard, with the lowest reported activation fractions on the order of five percent[38]. As opposed to most spectroscopic techniques and classical Laue time-resolved crystallography, in a time-resolved SFX experiment, each bR molecule is only ever exposed to one single femtosecond laser pulse, preventing the accumulation of light damage effects. Activated fractions furthermore represent a structural average throughout the crystal layers. To estimate this effect, we propagated the theoretical power densities throughout the crystal in thin slices of 45 Å (average distance of retinal chromophores) to monitor how the power density evolves within the crystal (Fig. 5A). Given an average laser path length of around 7 µm through our hexagonal microcrystals, the data collected at the lowest laser power density lies predominantly below the threshold reported as critical boundary in spectroscopic studies, while the data at the high end lie consistently above, aligning with the observation of the potential alternative photoproduct. Larger crystals are less sensitive to high laser power densities due to light attenuation throughout the crystal and typically diffract to higher resolution. Nevertheless, if smaller crystals diffract to sufficient resolutions, they are preferred as they allow for a more homogenous activation with lower laser pulse energy. The differences between solution and crystals are further highlighted by a recent laser power titration on bR crystals[27], where the authors suggest the use of peak fluences not greater than 100 mJ/cm², in good agreement with the onset of damage observed in our study (Supplementary Table 2). The study shows that high laser fluence increases the accumulation of early M-intermediate[27], which aligns with our results, where the laser-induced temperature jump could affect photocycle kinetics. Using spectroscopic data collected on crystals to guide a time-resolved crystallographic experiment could also make studies on challenging targets more feasible, as crystal supply is often limited, and the beamtime required to collect a single time point is usually longer due to lower hit rates. Nevertheless, the local power density in each unit cell throughout the crystal depends on many factors (Fig. 1A) and will be difficult to match exactly using an offline spectroscopic setup. In-line spectroscopy, as already applied for metalloproteins using X-ray spectroscopy[48], could potentially improve and verify outcomes directly.

Our findings suggest that functional photocycle intermediates are structurally observable at power densities that are far beyond what is called the linear regime in optical spectroscopy of bR in solution.

**A  Laser Penetration Depth**

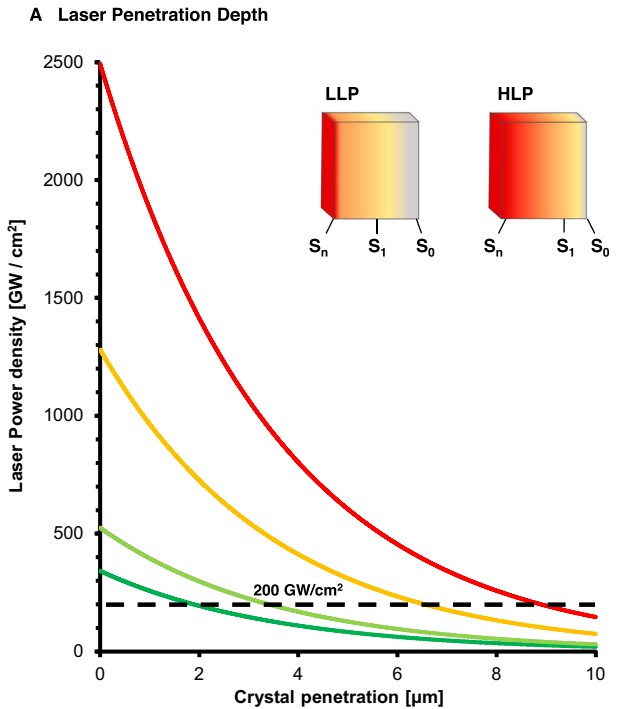

**B  Simplified Reaction Scheme**

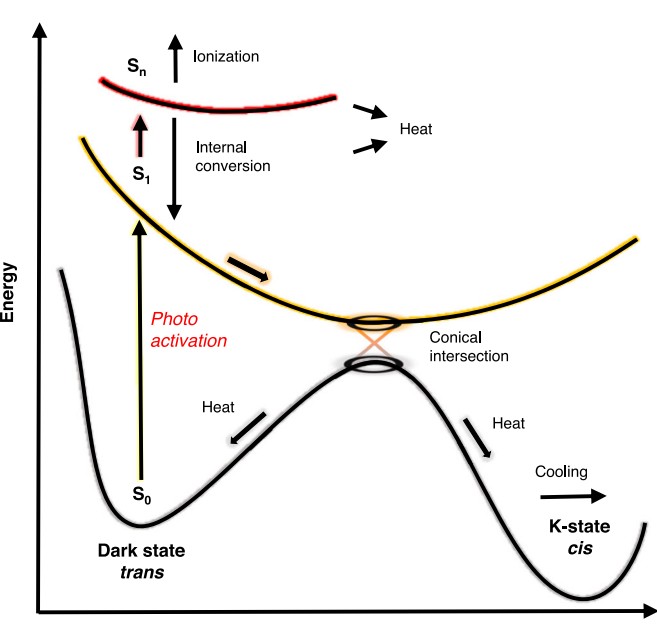

**Fig. 5 | Laser penetration and simplified reaction scheme. A** Estimated decay of laser power through bR crystals for the four femtosecond laser experiments. The inset shows a schematic representation of how dark (gray), $S_1$ (yellow), and $S_n$ (red) excited states populate within a crystal under low laser power (LLP) and high laser power (HLP), respectively. **B** Simplified potential energy diagram of retinal *trans* to *cis* isomerization within bR. Arrows indicate reaction pathways discussed throughout the manuscript.

Under the excitation conditions we used, the photocycle is expected to occur, as the fluence range examined by Engilberge et al.[27]. encompasses the entire range discussed here, and the signatures of later-stage intermediates are detected. Structural distortions that may have occurred due to the absorption of additional photons by the excited retinal state increase heterogeneity and reduce difference map quality but do not result in structural artifacts on the refined K-intermediate, except for the highest power density employed. Further experiments would have to show if structural dynamics remain consistent in the femtosecond range, where the retinal is still in the electronically excited state, or in the millisecond range, where cooling effects will have fully played out. Even though it is difficult to generalize to other systems because the photochemical parameters are specific for each reaction, it is important that we obtain the cleanest electron density maps with the lowest femtosecond laser excitation. This finding suggests that there is not necessarily a conflict between obtaining strong signals and staying close to the linear regime, overall reducing the need for hard choices when planning future experiments. With careful consideration of potential pitfalls, the perfect does not have to be the enemy of the good, and much of the biological 'music' of a reaction[49] can be preserved over a large excitation regime.

## Methods
### Protein purification and crystallization
Bacteriorhodopsin (UniProtKB P02945) purification and crystallization were conducted from purple membranes of *Halobacterium salinarum* under dim red light or in the dark as previously described in ref. 16. Briefly, the protein was solubilized from purple membranes by a 50 mM sodium phosphate buffer pH 6.9 supplemented with 1.7 % ß-octyl glucoside. After overnight incubation, the pH was adjusted to 5.5 with 0.1 M HCl, and the insoluble fraction was removed with a $150000 \times g$ ultra-centrifugation for 1 h. Size exclusion chromatography was avoided, according to ref. 11. Millipore centricolumns with 50 kDa

cutoff were used to concentrate bR up to 40–80 mg/ml. Lipidic cubic phase (LCP) was obtained by mixing the concentrated protein with monoolein at a ratio of 42:58 in Hamilton syringes.

Crystallization was set up in Hamilton syringes where LCP was incubated in a buffer containing 100 mM Sorensen buffer pH 5.6 and 30 % Polyethylene Glycol 2000. Syringes were incubated at 21 °C for 3-6 days. The obtained microcrystals were hexagonal plates around 20–35 µm in diameter and 2-3 µm in thickness, which results in an average laser path length through the crystal of ~7 µm as calculated according to ref. 29.

### Sample preparation
The crystallization buffer was removed from the LCP-containing crystals by slowly pressing out through the syringe coupler. Shortly before the experiment, monoolein was added to the sample until the obtention of a homogeneous and stable mesophase. The sample was then supplemented with 5% MAG 7.9 to prevent phase transition during injection, and 5% paraffin to obtain a smooth sample flow. The sample was homogenized using a custom-made 3-way coupler[50]. The sample was light-adapted at 300 mW for 5 min through a long yellow filter (> 515 nm) before data collection. The sample was then measured within the following 30 min after light exposure.

### Experimental setup SwissFEL experiment
The high and low laser power densities were recorded at SwissFEL[51] from the 8th to the 10th of December 2018 during the commissioning of the Alvra setup for TR-SFX experiments. The SwissFEL delivered X-ray pulses with a photon energy of 9 keV and a pulse energy of 300 µJ at a repetition rate of 25 Hz. The X-rays were delivered in 10–50 femtosecond pulses and were focused to a diameter of 5 µm. The air in the sample chamber was pumped down to 100–200 mbar while being substituted by helium, to reduce X-ray scattering.

Bacteriorhodopsin crystals were extruded through a 75 μm capillary with a high viscosity injector[52] connected to an HPLC pump at a speed of 9290 μm/s. The light-sensitive reaction was triggered at the X-ray interaction zone with a femtosecond laser, producing an 80 fs pulse at a wavelength of 533 nm. The spot size was 80 μm $1/e^2$ (50 μm FWHM), and the laser delivered energy of respectively 10.8 μJ and 1.43 μJ, the highest and lowest that were possible during commissioning.

A true dark dataset was collected to determine a dark structure and as control. The pump-probe experiment was done with alternating light-dark images or in a 4:1 light to dark image sequence where every fifth pulse of the laser was blocked. Each laser power dataset has a dark dataset recorded that way. These laser off images were compared to the true dark dataset to attest the absence of light contamination in each laser power setup.

### Experimental setup SACLA experiment

Data were recorded at SACLA on the 30th and 31st of July 2017 as one of the first experiments testing the femtosecond laser setup. The XFEL delivered photons with an energy of 10.1 keV with a total pulse energy of 622 μJ. The X-ray was focused to 1.5 μm. At the interaction point, the light-sensitive reaction was triggered by a laser with a wavelength of 531 nm and an energy of 13 μJ. The laser spot size was 140 μm $1/e^2$ (83 μm FWHM), and the laser pulse length was 70 fs. The sample was extruded through a 75 μm capillary at a speed of 9500 μm/s. Data were recorded with pump-probe delays of 10 ps, 100 ps 1 ns, and 10 ns. The XFEL pulses were used at a repetition rate of 30 Hz while the laser was triggered at 15 Hz, assuring that light and dark data were recorded in an interleaved manner. A true dark dataset was collected as a control for light contamination.

### Data processing

All data were processed using the CrystFEL 0.9.1 suite[53], peak finding was performed by peakfinder8 and indexing was done by Xgandalf[54], MOSFLM[55] and DirAx[56], with the options −threshold = 500 −int-radius = 3,4,7 −min-snr−4.2 −min-peaks = 8 for SwissFEL data and −threshold = 400 −int-radius = 3,4,7 −min-snr−3.5 −min-peaks = 8 for the SACLA data. While for data previously reported by Nango et al.[11]. only the 16 ns timepoint was analyzed in detail, we have re-indexed all timepoints in the nanosecond to millisecond range to determine the unit cell expansion. The indexing ambiguity inherent with the bR crystal space group was resolved using ambigator. Intensities were merged using partialator with the options -n 1 −push-res = 1.5 -m xsphere. Data statistics are summarized in Supplementary Table 3.

### Analysis of laser induced unit cell expansion

To analyze the differences between light and dark unit cells (Supplementary Table 2), we used the new data collected using femtosecond laser excitation at SACLA and the time series published in Nango et al.[11]. collected with a nanosecond laser of the same fluence. To account for variations in detector distance through the installation of the LCP injector, the average unit cell from each run was calculated separately. For each run, the dark averages of axes a, b, and c have been subtracted from the light averages, and the difference is then averaged over all runs of a timepoint. We calculated the volume of the unit cell according to the formula $V = 0.866$ a*b*c and normalized the differences to the average dark cell.

### Structure determination and refinement

The structure of bR was solved by molecular replacement using the dark model (pdb code: 6G7H) from Nogly et al.[16]. The retinal ligand RET, usually used for refinement of previous studies, was replaced by the ligand LYR composed by retinal and lysine 216. The LYR ligand constraints were defined with the GRADE2 server[57] and inspected manually. Refinement steps were performed using Phenix refine[58] and COOT[59]

iteratively. Structure quality was determined with MOLPROBITY[60]. Rmsd values were determined using CCP4 Lsqkab software[61].

### Difference B-factor analysis of light and dark datasets

To assess the effects of laser power density on crystallographically refined B-factors, we carried out refinements as follows. In the refined dark model, B-factors were randomized, and atomic positions were randomly shaken by 0.2 Å to generate ten different starting models. These models were then equilibrated over 100 cycles against all eight datasets (Supplementary Table 3) using reciprocal space and atomic, isotropic B-factor refinements. Data was anisotropically truncated by the Staraniso server[62], and the overall resolution was further truncated to 1.7 Å to balance differences in data quality. The atomic B-factors after refinement of the replicates were then averaged and normalized to the average refined B-factor of the model. A new model was generated containing the difference of the normalized B-factor for light and dark datasets in percent. These models were used to present a B-factor gradient in pymol for Fig. 2A and Supplementary Fig. 6.

### Difference electron density maps and data extrapolation

Difference electron density maps were generated by Phenix[58] and Xtrapol8[63] using the multiscaling method and reflections from the highest resolution available up to 5 Å as low-resolution cut-off. Phases were obtained from the bR dark model and refined against the respective dark dataset. Extrapolation was done by using CCP4[61] suite without weighting and with calculation following the formula $F_{ext}(light) = 100/A \times (F_{obs}(light) - F_{obs}(dark) + F_{obs}(dark))$. Where A is the activated fraction of molecules (for which the action has been triggered in the crystal) in percent, $F_{obs}(light)$ and $F_{obs}(dark)$ are the experimental structure factors for light and dark data, and $F_{ext}(light)$ is the extrapolated structure factor amplitude. The activation level or activated fraction of molecules was determined according to ref. 38 employing the integration of all negative $2F_{obs}-F_{cal}$ densities around retinal C20 with decreasing A values, using a script from ref. 64. When A is determined too low, negative $2F_{obs}-F_{calc}$ density appears in the maps, and the integrated negative density rises, indicating an undervaluation of the activation level. Datasets and refinement statistics are shown in Supplementary Table 1. Figures were made using PyMOL[65].

### Power density calculation

Every experiment analyzed in this study was performed with a Gaussian laser beam profile, and power densities were calculated accordingly. The formula used to calculate the Peak Fluence $F_0$ (in J/m²) was $F_0 = E/(\pi w^2)$ where E is the total energy of the laser pulse (in J) and $\pi w^2$ the laser surface area in m², calculated from w being the beam radius (half waist) at $1/e^2$. The peak power density $I_0$ was then calculated from T, the pulse length, as $I_0 = F_0/T$. The femtosecond experiments at SACLA/SwissFEL and the nanosecond experiment from Nango et al.[11]. experiments were performed at laser peak, so no offset had to be calculated. However, for Nogly et al.[16]. and Nass Kovac et al.[29]. an offset of 50 μm and 25 μm, respectively, was applied to the laser position. The peak power density was then calculated according to $I(x) = I_0 \cdot \exp(-2x^2/w^2)$ where x is the offset from the beam center, and I(x) is the laser power density at the offset assuming a Gaussian beam. The values used and calculated are shown in Supplementary Table 2.

### Reporting summary

Further information on research design is available in the Nature Portfolio Reporting Summary linked to this article.

## Data availability

Coordinates and structure factors for dark state bacteriorhodopsin have been deposited on the PDB database server under accession codes 9F9D (SwissFEL) and 9F9I (SACLA, fs laser). Models refined

against extrapolated data, extrapolated structure factors, and light data used for data extrapolation have been deposited under accession codes: 9F9E (342 GW/cm²), 9F9J (1281 GW/cm²), and 9F9F (2493 GW/cm²). Reprocessed dark and light data from[11] and the respective structures have been deposited under 9F9B (dark) and 9F9C (0.04 GW/cm²). Reprocessed dark and light data from[16] and the respective structures have been deposited under 9F9G (dark) and 9F9H (525 GW/cm²). The previously published PDB code used in this work is 6G7H. Source data for Figs. 2, 5, and Supplementary Fig. 6 are provided in a Source Data file. Source data are provided in this paper.

## Code availability

The code used to analyze the difference of B-factor between light and dark datasets in our SACLA, fs laser data is deposited on figshare (https://doi.org/10.6084/m9.figshare.27332034.v2).

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

## Acknowledgements

We are grateful for the excellent support from the PSI Crystallization Facility and the Macromolecular Crystallography group during the growing of crystals. We thank Robert Bosman and Takanori Nakane for supporting the data processing team and commenting on the manuscript. This project was founded by the following agencies: the Swiss National Science Foundation under projects grants 310030_197674 (to T.W.) and 310030_207462 (J.S.); and Swiss Innovation Agency Innosuisse grant 42711.1 IP-LS (to J.S.).

## Author contributions

Conceptualization: T.W., Q.B., and J.S. Sample preparation: P.N., A.F., and D.K. X-ray data collection: P.N., E.N., A.F., D.K., D.J., F.D., P.S., S.M., I.M., P.B., G.O., C.H., D.O., S.B., V.P., T.T., R.T., K.T., S.O., E.N., P.J.M.J., K.N., G.K., C.C., S.I., R.N., C.M., G.S., T.W., and J.S. X-ray data processing: Q.B., T.W., and G.K. Laser Power calculations: Q.B. and M.K. Writing - original draft: Q.B., T.W., and J.S. Writing - review and editing: All authors could read and comment on the manuscript.

## Competing interests

The authors declare no competing interests.

## Additional information

[1]Division of Biology and Chemistry, Paul Scherrer Institut, Villigen, Switzerland. [2]Institute of Molecular Biology and Biophysics, Department of Biology, ETH Zürich, Zürich, Switzerland. [3]RIKEN Spring-8 Center, 1-1-1 Kouto, Sayo-cho, Sayo-gun, Hyogo, Japan. [4]Japan Synchrotron Radiation Research Institute, 1-1-1 Kouto, Sayo-cho, Sayo-gun, Hyogo, Japan. [5]Department of Cell Biology, Graduate School of Medicine, Kyoto University, Yoshidakonoe-cho, Sakyo-ku, Kyoto, Japan. [6]Photon Science Division, Paul Scherrer Institut, Villigen, Switzerland. [7]Department of Chemistry and Molecular Biology, University of Gothenburg, Box 462, Gothenburg, Sweden. [8]Japan Science and Technology Agency (JST)–Precursory Research for Embryonic Science and Technology (PRESTO), 4-1-8 Honcho, Kawaguchi, Saitama, Japan. [9]Present address: Dioscuri Center for Structural Dynamics of Receptors, Faculty of Biochemistry, Biophysics and Biotechnology, Jagiellonian University, Gronostajowa 7, Krakow, Poland. ✉e-mail: tobias.weinert@psi.ch; joerg.standfuss@psi.ch

