## [Transparent Peer Review file · Nature Communications]

Structural effects of high laser power densities on an early bacteriorhodopsin photocycle intermediate

Corresponding Author: Dr Jörg Standfuss

Version 0:

Reviewer comments:

Reviewer #1

(Remarks to the Author)

The quantitative effects of high laser power density on time-resolved serial crystallographic structures have been largely unknown. The authors have undertaken determining the impact of high laser power densities on serial crystallographic data which needs to be thoroughly investigated to improve accuracy of photochemical interpretation by the many new investigators undertaking serial crystallography. They expertly compare light-activation effects within crystals compared to that in solution and analyze and measure the impact they observed as changes on the bacteriorhodopsin atomic structure of the K intermediate, an early, red-shifted intermediate which they find do produce potential atomic conformational changes. This uncertainty of changes are specific to high intensity laser light that may affect the chemistry, however, importantly, the changes are subtle and K can be tracked. They find that the K retinal and interacting opsin structure are slightly altered with high photoexcitation values, but the changes are subtle.

An important outcome of this study is to quantitate and alert new users that actinic light intensities need to be monitored, controlled, and preferably minimized to the values sufficient to measure specific changes in the photochemical reaction cycle. The authors are experts as is evident in their data and their analysis.

Reviewer #2

(Remarks to the Author)

The article "Structural effects of high laser power densities on an early bacteriorhodopsin photocycle intermediate" by Bertrand et al. presents a selection of pump-probe time-resolved serial crystallographic data of bacteriorhodopsin including new and reprocessed data from previous studies. The different datasets include a variation of the laser power density ranging from 0.04 to 2493 GW/cm², of the time delay ranging from 10 ps to 1.7 ms, as well as the use of a femtosecond or nanosecond pump laser. Based on their refinement, the authors study different effects, local and global, induced by the various combinations: B-factors, cell volumes and electron density map variations.

This study contributes to the current debate in the serial crystallography field about the appropriate pumping laser parameters for time-resolved experiments. It clearly presents the effect of a high laser power density on the bacteriorhodopsin structure and compares the results to other studies in solution.

The result is strong manuscript with a well considered argument on the choice and effects of laser fluence using a system well chosen to be sensitive to the relevant effects. This study is interesting for the field and thus I recommend it for publication with minor revisions (see comments below).

Overall remarks:

In the discussion, the authors could also comment on the effect of crystal size and morphology in relation to their findings regarding appropriate laser fluence. What about larger or smaller crystals than used in the study? What advice based on your findings can you give regarding the laser power necessary for a decreasing crystal size?

Figure 1B: The bR states and time scales are not clearly represented. This could be briefly described so that the studied K-state is clear even for non bR experts.

Figures 3 and 4: The “light green” model presented in the 342 GW/cm² panels is not listed in the legend. It would also be easier for the readers to have a representation of the reference 0.04 GW/cm² maps for direct comparison, rather than a reference to the supplementary figure in the main text.

Line 463: Can the authors confirm that, especially in the previous studies, the laser beams were Gaussian? This would affect the comparative calculation in the case that other studies had used a ‘top-hat’ beam for example.

Comments / Typos etc.:

Line 125: “returns” instead of “returned” for tense consistency.

Line 259: There is a missing “of” in the part “The question how”.

Line 295: “the” in front of “nano” should be removed.

Line 319: It could be made clearer by adding “in a time resolved SFX experiment” before “each bR molecule”.

Line 324: This line should refer to Figure 5A and not 5B. Figure 5B is then not mentioned at all in the main text.

Line 398: There is a missing “at” in the part “capillary a speed of”.

Line 452: There is a missing “)” in the formula.

Line 466: There is a space missing after “ πw^2 ”.

Line 471: There is an additional space before both “ μm ”.

References 19 and 30 (cited lines 84 and 112) as well as 40 and 51 (cited lines 200 and 318) are the same, please correct and adjust numbers accordingly.

Supplementary Figure 5: There is a typo in panel C label “Fobs(trued-ark)” should be “Fobs(true-dark)”.

Reviewer #3

(Remarks to the Author)

Review of Structural effects of high laser power densities on an early bacteriorhodopsin photocycle intermediate
NCOMMS-24-33658

Summary

This manuscript by Bertrand et al continues the investigation of the effect of laser power on light activatable samples when triggering reactions for time-resolved X-ray crystallography. This topic has been at the center of many discussions within the community over the last decade, with more and more emphasis being placed on correct light-triggering conditions for photoactive systems as to not overinterpret information obtained from data that may be overexposed or suffer from non-linear effects that distort the biologically relevant physical mechanisms that govern these types of systems. It is now known and accepted that higher laser powers can cause such effects, the question here is how much more this research adds to the field of study. The main takeaway from the publication is that structurally big changes are not seen from increased laser power. But, as stated, the calculated laser powers suffer from severe caveats, and thus the quoted values can be up to 90% off the actual power density the crystals receive.

Specific comments

In the introduction, the authors point at analyzing and comparing TR-SFX data of the target protein under varying illumination conditions to reach the best balance between activation and photon damage. Though true, such work is prohibitive for samples other than those that are extremely readily available and already well established. Available beamtime for such an experiment is also scarce, and thus the community will generally focus on performing other, less demanding experiments, such as simpler power titrations with spectroscopy to determine their optimal light activation conditions.

The authors point at all the previous work done with spectroscopy that already indicates the formation of different intermediates depending on photon flux for laser excitation and highlight that structural information from these regimes is lacking. The question remains of whether collecting data at these regimes is even suitable to begin with if we know from spectroscopic data that the system isn't progressing through a biologically relevant pathway? Maybe no specific, short-lived structural changes are visible for this system, but how generalizable is this idea to other systems or longer timepoints?

The main takeaway from the publication seems to be the correlation between multiphoton absorption at a fs and ns scale and its influence at longer timescales and later structural intermediates.

The approximations taken to compare the datasets do not consider light scattering, which according to literature is a very large source of change in photon density, so how can these datasets be comparable to an extent that is significant?

In figure 3 – the comparison of the different structures collected at different laser powers. The maps clearly show differences, but it is unclear whether these differences are truly just from the effect of the laser. In the main text, the authors clearly state that the datasets have different multiplicities and time resolutions, do they also have different resolutions? The increasing laser power shows a possible change, but there is no clear trend of structural change, so I am not convinced that these changes can be solely attributed to the laser power used.

Supplementary figure 7 clearly shows a dependency of the extrapolated maps and structures not only on laser power density, but in other factors such as data resolution and activation level, further shedding doubt into the comparability of the datasets in this study.

The changes highlighted in the data comparison in figure 3, obtained by the Pandey et al method effectively shows one

dataset changing, the highest laser power – could this just be an artifact of this one dataset? For robust analysis, an even higher dose dataset would be needed to see the continuing trend.

For clear understanding of whether the changes observed are due purely to laser power or initiated transitions, the authors must specify whether at any of these laser powers protein photocycle should occur. Furthermore, in the datasets being compared, there is no indication of whether the time delay is the same (figures 4 and 3) or if they correspond to different time delays.

Figure 5 shows a diagram for the expected penetration of the laser in the sample at different laser power densities, though nowhere in the text is there a description of what the crystals sizes used in the different experiments were, so it is unclear how significant this information is.

In terms of the conclusions and discussions of findings, the first general advice given is not related to any of the work presented, but just a general rule of thumb already adopted by the community for these experiments. This fact decreases the impact of the paper significantly.

Further conclusions highlight that the laser power at which structural changes are observed is much higher than those that have been reported to cause spectroscopic changes in solution. What about spectroscopic changes in the crystals? Do these correlate to the structural changes observed? This discrepancy is further highlighted by the highlight that the laser power at the sample may be significantly decreased due to scattering, and thus the conclusions being drawn based on the laser power described do not actually correspond to the number of photons that are absorbed at the sample.

The final main conclusion by the authors is that structurally, the protein is much less sensitive to changes from increased laser powers in crystals than spectroscopically in solution. Can the authors comment on the relevance of this assertion? Is this a crystalline effect? Shouldn't the comparison be with spectroscopy in crystals? Is there actually changes that happen at longer timescales from multiphoton absorption not commented on in this work?

Minor suggestion

It is unclear what the correlation between the nanosecond laser data and the femtosecond laser data should be in figure 3. Please clarify in the text for the non-expert.

Reviewer #4

(Remarks to the Author)

Version 1:

Reviewer comments:

Reviewer #2

(Remarks to the Author)

I am satisfied that the authors have fully considered the feedback contained in the initial review round and made relevant updates to the manuscript. The manuscript is improved as a result and I would recommend publication at this stage.

Reviewer #3

(Remarks to the Author)

I thank the authors for a thorough response to the comments and revision of sentences to make points clear to members of the community that are interested in laser-activated TR-MX but who are not deep in the TR bacteriorhodopsin field as well as addressing the uncertainties highlighted. With the revisions made, I recommend it for publication.

Reviewer #4

(Remarks to the Author)

REVIEWER COMMENTS

Reviewer #1 (Remarks to the Author):

The quantitative effects of high laser power density on time-resolved serial crystallographic structures have been largely unknown. The authors have undertaken determining the impact of high laser power densities on serial crystallographic data which needs to be thoroughly investigated to improve accuracy of photochemical interpretation by the many new investigators undertaking serial crystallography. They expertly compare light-activation effects within crystals compared to that in solution and analyze and measure the impact they observed as changes on the bacteriorhodopsin atomic structure of the K intermediate, an early, red-shifted intermediate which they find do produce potential atomic conformational changes. This uncertainty of changes are specific to high intensity laser light that may affect the chemistry, however, importantly, the changes are subtle and K can be tracked. They find that the K retinal and interacting opsin structure are slightly altered with high photoexcitation values, but the changes are subtle.

An important outcome of this study is to quantitate and alert new users that actinic light intensities need to be monitored, controlled, and preferably minimized to the values sufficient to measure specific changes in the photochemical reaction cycle. The authors are experts as is evident in their data and their analysis.

We thank the reviewer for the encouraging assessment of our work.

Reviewer #2 (Remarks to the Author):

The article “Structural effects of high laser power densities on an early bacteriorhodopsin photocycle intermediate” by Bertrand et al. presents a selection of pump-probe time-resolved serial crystallographic data of bacteriorhodopsin including new and reprocessed data from previous studies. The different datasets include a variation of the laser power density ranging from 0.04 to 2493 GW/cm², of the time delay ranging from 10 ps to 1.7 ms, as well as the use of a femtosecond or nanosecond pump laser. Based on their refinement, the authors study different effects, local and global, induced by the various combinations: B-factors, cell volumes and electron density map variations.

This study contributes to the current debate in the serial crystallography field about the appropriate pumping laser parameters for time-resolved experiments. It clearly presents the effect of a high laser power density on the bacteriorhodopsin structure and compares the results to other studies in solution.

The result is strong manuscript with a well considered argument on the choice and effects of laser fluence using a system well chosen to be sensitive to the relevant effects. This study is interesting for the field and thus I recommend it for publication with minor revisions (see comments below).

We thank the reviewer for the encouraging assessment and the contribution to improve the manuscript. The minor revisions requested by the reviewer have been implemented as detailed below.

Overall remarks:

In the discussion, the authors could also comment on the effect of crystal size and morphology in relation to their findings regarding appropriate laser fluence. What about larger or smaller crystals than used in the study? What advice based on your findings can you give regarding the laser power necessary for a decreasing crystal size?

The size and morphology of crystals indeed influence the required laser fluence. The smallest crystals that yield the resolution necessary to observe the structural changes that occur, at the smallest laser power that yields sufficient activation to observe the changes and refine a model against extrapolated data would be ideal. As shown in Figure 5A, decreasing crystal size should necessitate a reduction in laser power due to lower average attenuation within the crystal. We believe that ultimately determining the optimal laser power can only be done experimentally, due to the many variables influencing the experiment. While conclusions based on spectroscopic data are informative, they may not necessarily provide the whole picture. That said, spectroscopy done on bR crystals does show a similar condition, suggesting that a workflow combining spectroscopy on crystals, with a small power titration, around the spectroscopically defined boundary of an early time point (revealing a well-defined kinetic state) could improve results. Such an approach is supported by our data, as the best difference maps were observed at the lower end of the laser power in agreement with the spectroscopy on bR crystals presented in Engilberge et al.¹. We have adapted the paragraph below to discuss the effect of laser power and crystal size.

“Given an average laser path length of around 7 μm through our hexagonal microcrystals, the data collected at the lowest laser power density lies predominantly below the threshold reported as critical boundary in spectroscopic studies, while the data at the high end lie consistently above, aligning with the observation of the potential alternative photoproduct.”

Larger crystals are less sensitive to high laser power densities due to light attenuation throughout the crystal and may typically diffract to higher resolution. Nevertheless, if smaller crystals diffract at sufficient resolution, they are preferred as they allow for a more homogenous activation with lower laser pulse energy. The differences between solution and crystals are further highlighted by a recent laser power titration on bR microcrystals¹ where the authors suggest to use peak fluences not greater than 100 mJ/cm², in good agreement with the onset of damage observed in our study (**Supplementary Table 2**). The study shows that high laser fluence increases the accumulation of early M-intermediate¹, which aligns with our results, where the laser-induced temperature jump could affect photocycle kinetics. Using spectroscopic data collected on crystals to guide a time-resolved crystallographic experiment could also make studies on challenging targets more feasible, as crystal supply is often limited and the beamtime required to collect a single time point is usually longer due to lower hit rates. Nevertheless, the local laser power density in each unit cell throughout the crystal depends on many factors (**Figure 1A**) and will be difficult to match exactly using an offline spectroscopic setup. In-line spectroscopy, as already applied for metalloproteins using X-ray spectroscopy², could potentially improve and verify outcomes directly.”

Figure 1B: The bR states and time scales are not clearly represented. This could be briefly described so that the studied K-state is clear even for non bR experts.

The photocycle in Figure 1B has been updated to be more readable. The text of the caption has also been improved with the sentence:

“All datasets discussed in our study were recorded at a time delay where bR is expected to be in the K-intermediate. The next intermediate, L, appears later in the photocycle, and becomes visible in datasets recorded at 290 ns³.”

Figures 3 and 4: The “light green” model presented in the 342 GW/cm² panels is not listed in the legend. It would also be easier for the readers to have a representation of the reference 0.04 GW/cm² maps for direct comparison, rather than a reference to the supplementary figure in the main text.

This is correct. We have now added a few words in the captions of Figure 3 and 4 that link the light green model to the 342 GW/cm² datapoint. The decision to remove the 0.04 GW/cm² maps from Figure 3 and 4 was difficult, but we chose to do so to allocate more space for the other four maps and better highlight the phenomena observed using more common femtosecond lasers, which are discussed in greater details along the article.

Line 463: Can the authors confirm that, especially in the previous studies, the laser beams were Gaussian? This would affect the comparative calculation in the case that other studies had used a ‘top-hat’ beam for example.

We confirm to the reviewer that the laser beams were always Gaussian, due to the optical setup used in the experiments. To highlight this, we changed the sentence in Material and Methods to:

“Every experiment analyzed in this study was performed with a gaussian laser beam profile and power densities were calculated accordingly.”

Comments / Typos etc.:

Line 125: “returns” instead of “returned” for tense consistency.

Line 259: There is a missing “of” in the part “The question how”.

Line 295: “the” in front of “nano” should be removed.

Line 319: It could be made clearer by adding “in a time resolved SFX experiment” before “each bR molecule”.

Line 324: This line should refer to Figure 5A and not 5B. Figure 5B is then not mentioned at all in the main text.

Line 398: There is a missing “at” in the part “capillary a speed of”.

Line 452: There is a missing “)” in the formula.

Line 466: There is a space missing after “ πw^2 ”.

Line 471: There is an additional space before both “ μm ”.

References 19 and 30 (cited lines 84 and 112) as well as 40 and 51 (cited lines 200 and 318) are the same, please correct and adjust numbers accordingly.

Supplementary Figure 5: There is a typo in panel C label “Fobs(trued-ark)” should be “Fobs(true-dark)”.

We thank the reviewer 2 for the careful reading of the article. All suggestions have been implemented into the revised version of the manuscript.

Reviewer #3 (Remarks to the Author):

Review of Structural effects of high laser power densities on an early bacteriorhodopsin photocycle intermediate
NCOMMS-24-33658

Summary

This manuscript by Bertrand et al continues the investigation of the effect of laser power on light activatable samples when triggering reactions for time-resolved X-ray crystallography. This topic has been at the center of many discussions within the community over the last decade, with more and more emphasis being placed on correct light-triggering conditions for photoactive systems as to not overinterpret information obtained from data that may be overexposed or suffer from non-linear effects that distort the biologically relevant physical mechanisms that govern these types of systems. It is now known and accepted that higher laser powers can cause such effects, the question here is how much more this research adds to the field of study. The main takeaway from the publication is that structurally big changes are not seen from increased laser power. But, as stated, the calculated laser powers suffer from severe caveats, and thus the quoted values can be up to 90% off the actual power density the crystals receive.

We appreciate the reviewer's time and thoughtful comments on our article. It is true there have been many discussions within the last decade but there is precious little known about what the photodamage actually looks like on the structural side. As also stated by the other reviewers our work fills this important gap using a much-discussed system. We further believe there may have been a misunderstanding regarding the level of uncertainty associated with the measured laser power density in the experiments we analyzed. One of the main points of our study was that the local laser power densities in a TR-SFX experiment depend on numerous factors and cannot be stated in absolute terms, as it is sometimes done when discussing multi-photon effects. Instead, we use only well-known experimental parameters and state absolute laser power densities for our comparisons, while pointing out throughout the manuscript that the calculation of the absolute number of photons each bR molecule receives is largely futile and prone to error. This includes the quoted values of up to 99% due to the delivery medium.

We would like to point out that, in the text, we refer to calculations where experimental conditions can lead to a calculated decrease of effective laser power up to 99 % when opaque media are used for extrusion⁴. However, the media we used was LCP (for each experiment) which is mostly transparent. Therefore, we estimate the amount of scattering to be between 20%⁵ and 35%⁶.

To clarify this point, we added this sentence:

“For LCP, the medium used for data collection of each experiment we analyzed, scattering contributions of 20%⁵ and 35%⁶ have been experimentally determined.”

Specific comments

In the introduction, the authors point at analyzing and comparing TR-SFX data of the target protein under varying illumination conditions to reach the best balance between activation and photon damage. Though true, such work is prohibitive for samples other than those that are extremely readily available and already well established. Available beamtime for such an experiment is also scarce, and thus the community will generally focus on performing other, less demanding experiments, such as simpler power titrations with spectroscopy to determine their optimal light activation conditions.

This is correct, we hope that incorporating spectroscopy on crystal as a routine experiment will help mitigate the challenges of relating spectroscopic measurement to crystallographic experiment, while still providing important insights into sample behavior under strong laser light. We added the following sentence to address the reviewer's point:

“The study shows that high laser fluence increases the accumulation of early M-intermediate¹, which aligns with our results, where the laser-induced temperature jump could affect photocycle kinetics. Using spectroscopic data collected on crystals to guide a time-resolved crystallographic experiment could also make studies on challenging targets more feasible, as crystal supply is often limited and the beamtime required to collect a single time point is usually longer due to lower hit rates. Nevertheless, the local power density in each unit cell throughout the crystal depends on many factors (Figure 1A) and will be difficult to match exactly using an offline spectroscopic setup. In-line spectroscopy, as already applied for metalloproteins using X-ray spectroscopy², could potentially improve and verify outcomes directly.”

The authors point at all the previous work done with spectroscopy that already indicates the formation of different intermediates depending on photon flux for laser excitation and highlight that structural information from these regimes is lacking. The question remains of whether collecting data at these regimes is even suitable to begin with if we know from spectroscopic data that the system isn't progressing through a biologically relevant pathway?

We believe the TR-SX field has been hindered for several years by concerns about artefacts that have not yet been structurally visualized or categorized. For this reason, studying the evolution of difference densities and refined models with increasing laser power densities is crucial for advancing the field of time-resolved crystallography. Most spectroscopic methods are very sensitive and may overstate the actual molecular impact observed during a TR-SX experiment. Providing structures obtained with varying laser excitations is an important contribution to be able to quantify these changes and to learn to what extent they may compromise biological behavior.

In a way this is similar to the old question to what extent X-ray radiation damage compromises crystallographic structures. It is now well established that high X-ray doses break cysteine disulfide bonds and the presence of such artifacts clearly indicates to experimenters that a high X-ray dose was delivered to the crystals. This understanding was achieved through systematic studies of radiation damage on protein crystals. Importantly, even in cases of radiation damage we now know that it does not necessarily compromise the biological relevance of the structure. Therefore, we believe that aspects of photoexcitation should be further studied and that evidence from pump-probe experiments with high laser powers should not be disregarded, especially when there is evidence from other experiments that supports such findings.

Our findings demonstrate that the structural states obtained in bR over a broad range of laser power densities are generally robust, and that the excess energy is dissipated as heat. This suggests that the structural states that accumulate are biologically relevant and can provide insight into key residues. However, we also show that lower power densities lead to cleaner difference electron density maps, while excessive laser power increases heterogeneity - a valuable information that will guide future experiments.

Maybe no specific, short-lived structural changes are visible for this system, but how generalizable is this idea to other systems or longer timepoints?

We anticipate that proteins similar to bR, such as other retinal-binding bacterial light-driven pumps, will exhibit similar behavior under high laser power densities. Spectroscopic studies have demonstrated that, for bR, later intermediates may begin to accumulate at earlier time points when subjected to high power densities, while the characteristics of these later intermediates remains unchanged¹. The findings reported by Engilberge *et al.* align with both the heat dissipation we observe (Figure 2A), which speeds up transition kinetics, and the alternate retinal conformation we observe, which may resemble the L-intermediate.

In contrast, the robustness to initial excitation conditions observed in bR cannot be generalized to other proteins. However, we believe a more general principle can be drawn: calculated laser powers do not always reflect the actual laser power witnessed by the average unit cell within the crystal. Careful data analysis, taking other experimental data into account, will enable more informed decisions making. To address the referee's concerns, we have further highlighted the importance of the phytochrome study previously mentioned (Do *et al.*) (line 276):

“Spectroscopic studies on phytochrome show significant effects of high laser power densities⁷, pointing out that the chemical nature of the ligand (e.g. ionization energy) must always be considered, and that different systems react differently to high laser power activation.”

The main takeaway from the publication seems to be the correlation between multiphoton absorption at a fs and ns scale and its influence at longer timescales and later structural intermediates.

The approximations taken to compare the datasets do not consider light scattering, which according to literature is a very large source of change in photon density, so how can these datasets be comparable to an extent that is significant?

As previously discussed, we used experimental data obtained with the same LCP media under very similar conditions, meaning that the extent of scattering and the effects illustrated in Figure 1A are comparable. One of the main points of our article is that effects are best compared in relation to one another, using the applied laser power as a reference but not as an absolute measure.

To address the uncertainty associated with calculated laser powers, we have added a sentence in the discussion:

“Since laser scattering is reported with various strength and calculated differently, we excluded light scattering effects from the reported power densities. Instead, we relied solely on experimentally measured laser parameters for a more accurate comparison. In our study, all datasets were recorded with similar experimental setups, which allows us to consider the difference in laser scattering in between experiments as minimal.”

In figure 3 – the comparison of the different structures collected at different laser powers. The maps clearly show differences, but it is unclear whether these differences are truly just from the effect of the laser. In the main text, the authors clearly state that the datasets have different multiplicities and time resolutions, do they also have different resolutions? The increasing laser power shows a possible change, but there is no clear trend of structural change, so I am not convinced that these changes can be solely attributed to the laser power used.

The reviewer raises a point we have discussed much during preparation of the manuscript. While all datasets are of high quality, they do vary in resolution depending on which X-ray detector was used for data collection and on the amount of data that was collected. It would have been an option to truncate all datasets to the same resolution, however we did not want to “throw away” data. Instead, we made Supplementary figure 1, where all datasets were cut at the same resolution (1.8 Å) for a better comparison. From this figure it is evident that even when cutting the resolution, we observe the same difference map features dependent on laser power.

Furthermore, the absence of distinct structural changes across a wide range of laser power densities is discussed in the paper. The data indicates that high laser power does not disrupt the overall protein structure; instead, it increases heterogeneity. The reviewer should also consider that the SwissFEL data (at both high and low laser power density) were collected during the same beamtime using the same setup and sample, making them directly comparable and empathizing this trend. Therefore, we respectfully disagree with the reviewer’s statement, as this trend is observable in Figure 3 and Supplementary Figure 1.

Supplementary figure 7 clearly shows a dependency of the extrapolated maps and structures not only on laser power density, but in other factors such as data resolution and activation level, further shedding doubt into the comparability of the datasets in this study.

While the caption of Supplementary Figure 7 aimed to clarify the complexity of the extrapolated maps and structures, it seems to have raised concerns that we would like to address in more detail.

Supplementary Figure 7 illustrates an interesting phenomenon observed in our time-resolved experiments, when activation levels are determined by retinal motion. Specifically, the negative difference density on retinal C20 indicates retinal movement into the K-intermediate, and the Pandey *et al.*⁸ method assumes that the extrapolated density at this position should become zero when the correct multiplication factor is applied.

At elevated laser power densities, an additional complexity arises: the negative difference density observed is partly due to an alternative retinal conformation and the presence of "hot" retinal. In this state, some of the hot retinal transitions into the K-intermediate, while other parts fall back to the ground state, releasing heat that is transferred to nearby water molecules. This heat increases their vibrational disorder, which manifests as a negative peak on the water molecules. The directed motion of Water 402 is part of the biological photocycle process and occurs only during productive isomerization. It is therefore also visible at low power densities, showing a negative difference map peak on the side of Water 402, since the shift is smaller than the atom diameter.

However, because the heat transferred to the water molecules also comes from other processes (e.g., hot retinal returning to the ground state), the negative difference density associated with water molecules differs from that seen on retinal C20. Retinal C20 is relatively fixed, so even if "hot," its vibrational motion is less pronounced than that of water molecules. As a result, applying the multiplication factor derived from retinal isomerization leads to an overestimation of the water component, especially when the activation level is low (the multiplier is high). This can diminish the observed extrapolated density.

This is further complicated by the fact that these small vibrational motions are more visible at higher resolutions, meaning that the resolution of the data plays a role in how well this motion is captured. In contrast, difference maps, which do not rely on extrapolation, clearly show the increased heat signature in the water cluster at higher laser power densities (Supplementary Figure 1).

The purpose of Supplementary Figure 7 was to demonstrate this complex interplay of heat signal, resolution, and activation level determination. We have revised the caption to provide a more in-depth explanation for the interested reader.:

“The extrapolated maps illustrate the combined effect of laser power density, spatial resolution, and activation level on the observed density in the water cluster near the retinal. Higher laser power densities (C-E) enhance the negative difference density on the water molecules seen in Supplementary Figure 1, especially on Water 401, as heat is transferred during retinal isomerization and ground-state relaxation processes. At the same time, higher resolution allows for better visualization of subtle shifts in the water molecule positions, contributing to a more pronounced difference density. In panel (C), despite the lower laser power density, the higher spatial resolution and lower activation level (determined by retinal motion) lead to increased sensitivity to positional shifts in the water molecules. The higher resolution better captures the small vibrational displacements, resulting in lower observed extrapolated density – an “overextrapolation”. In panel (E), the highest laser power density, combined with a higher activation level that includes an additional retinal conformation, reduces the “overextrapolation” effect. This figure demonstrates the complex interplay between heat transfer, spatial resolution, and activation determination, showing how each factor influences the observed extrapolated density in the water cluster.”

The changes highlighted in the data comparison in figure 3, obtained by the Pandey et al method effectively shows one dataset changing, the highest laser power – could this just be an artifact of this one dataset? For robust analysis, an even higher dose dataset would be needed to see the continuing trend.

The high laser power dataset was recorded during the same beamtime as the low laser power dataset. This means that they share the same laser setup, crystals, media, injector/injection setup, detector, and X-ray beam properties. Therefore, it is highly unlikely that an artifact present in one dataset would be absent from the other, given that the method used is serial crystallography and data are recorded from tens of thousands of crystals.

The change in the observed activation level further suggests the accumulation of a different structural state. We were very careful in both assessing the relevance of the observed state and analyzing the data. Subtle hints in the SACLA data indicate a trend, which is also reflected in the slope of the curves used for activation level determination, however, we found this evidence insufficiently strong to present in the current analysis. We agree that including datasets with higher and lower laser power densities would enhance the clarity of the study. Unfortunately, this was not possible at the time of data collection where we used the highest and lowest settings available at SwissFEL. Even though more choices are now, our data do show a trend important to the community. As pointed out by the referee limited access to XFELs makes it not possible to collect more data with even higher power densities.

For clear understanding of whether the changes observed are due purely to laser power or initiated transitions, the authors must specify whether at any of these laser powers protein photocycle should occur.

The photocycle is expected to occur under these excitation conditions, as the fluence range examined by Engilberge *et al.*¹ encompasses the entire range discussed here, and the signature of later-stage intermediates are detected. Furthermore, our observed

difference map at 10 ns (obtained at intermediate laser power density) compare well with the 16 ns map recorded by Nango *et al.*³ at the same fluence, where later-stage intermediates are also observed. The structures are also very similar to those of the K-state determined using freeze-trapping experiments (for a comparison of both techniques see Wickstrand, C., et al. Annu Rev Biochem, 2019). Therefore, we are confident that the photocycle occurs under these illumination conditions.

We have included a sentence in the discussion to point out that we are certain that the photocycle will occur under the conditions used.

*“Under the excitation conditions we used, the photocycle is expected to occur, as the fluence range examined by Engilberge *et al.*¹ encompasses the entire range discussed here, and the signatures of later-stage intermediates are detected.”*

Furthermore, in the datasets being compared, there is no indication of whether the time delay is the same (figures 4 and 3) or if they correspond to different time delays.

The time delay for each experiment presented in the article is shown in Figure 1B. All experiments shown in Figures 3 and 4 are recorded at 6 ps for SwissFEL and 10 ps for LCLS and SACLA. At these time points, bR is in its K-intermediate with the next intermediate only occurring after hundreds of ns.

This is mentioned in the text, in lines 123-124 *“Here, we investigate the effects of increasing laser power density on the light-driven proton pump bacteriorhodopsin with a focus on the K-intermediate”* and in lines 160-161 *“The overall structures are very similar in the temporal range from 6 ps to 16 ns where the K-intermediate of bR is dominant”*. Since the intermediate occurring on these time scales is the same, both according to spectroscopy and our data, our focus was on the intermediate rather than the time delay. To address the referee’s concern, we have mentioned the K-intermediate in the figure titles and added the time delays for each experiment additionally to Figure 3 and 4 captions.

Figure 5 shows a diagram for the expected penetration of the laser in the sample at different laser power densities, though nowhere in the text is there a description of what the crystals sizes used in the different experiments were, so it is unclear how significant this information is.

The crystal size is mentioned in the “Materials and methods” section, under the “Protein purification and crystallization”, where it is stated that “The obtained microcrystals were hexagonal plates around 20-35 μm in diameter and 2-3 μm in thickness,...” We have further mentioned the average laser path lengths and crystal size in the main text as detailed below.

“Given an average laser path length of around 7 μm through our hexagonal microcrystals, the data collected at the lowest laser power density lies...”

We have now included the average path length in the caption of Figure 5 for easier accessibility.

“The average path length in the crystals used was 7 μm .”

In terms of the conclusions and discussions of findings, the first general advice given is not related to any of the work presented, but just a general rule of thumb already adopted

by the community for these experiments. This fact decreases the impact of the paper significantly.

We agree that the guideline ‘stay away from ESA’ is a widely adopted general rule, and that is the intent behind our statement in the manuscript. Importantly, we wish to highlight that despite adhering to this guideline, we did observe effects that likely result from the absorption of additional photons, probably absorbed by the excited state.

To avoid misunderstandings and reduce emphasis, we have changed the text as shown below:

“It is generally an adapted practice to ‘Stay away from ESA’, i.e. minimize the overlap between excitation wavelength and excited state absorption (ESA). This simple rule of thumb has helped minimizing direct multiphoton effects from using femtosecond lasers. However, at the highest laser power densities in our bR experiments, this was not sufficient to prevent the direct deposition of heat or its formation through internal conversion between S_n and S_1 excited states (Figure 5A & B).”

However, we strongly disagree with the statement that suggests our findings are not significant. Our study is the first to investigate the actual structural effects of photodamage in bacteriorhodopsin, which has long served as a model system in photobiology. As such, it holds substantial interest for many readers.

We also observe heat dissipation effects and the impact of increased heterogeneity on difference maps, which are relevant in other laser-triggered time-resolved experiments. Our findings indicate that, despite the increasing heterogeneity in the difference map signals, the refined structures remain similar—an outcome of particular interest to the TR-SFX community. While we acknowledge that completely general conclusions cannot be drawn, our analysis will aid the serial crystallography community in better understanding the nature and influence of photodamage effects. As already mentioned above, there were decade long discussions on what these effects are and here we have resolved them on the molecular level. We see our work as an ideal fit for the collection “Free-electron lasers — development and application” as suggested during the initial submission.

We intentionally refrain from providing bold recommendations on how experiments should be conducted, as we believe that scientists undertaking these experiments need information rather than prescriptive guidelines, which is what we aim to provide.

Further conclusions highlight that the laser power at which structural changes are observed is much higher than those that have been reported to cause spectroscopic changes in solution. What about spectroscopic changes in the crystals? Do these correlate to the structural changes observed? This discrepancy is further highlighted by the highlight that the laser power at the sample may be significantly decreased due to scattering, and thus the conclusions being drawn based on the laser power described do not actually correspond to the number of photons that are absorbed at the sample.

Recent spectroscopic experiments by Engilberge *et al.*¹ investigate the effect of laser fluence on bR crystals, identifying three distinct reaction populations depending on the laser power used. This study provides valuable insights into the spectroscopic effect of laser power on bR crystals.

The authors examined the effect of laser power on the M state population, with their setup allowing observation on time scale of 3 μ s, in the shortest time range, allowing visualization of later photocycle states of bR. The authors determined 3 groups based on

laser power effects. For group 1: below $75 \mu\text{J}\cdot\text{cm}^{-2}$, the profile of M-intermediate evolution resembled that recorded by Efremov *et al.*⁹, which has been recorded with the lowest laser power. For group 2 and group 3, between $81 \mu\text{J}\cdot\text{cm}^{-2}$ and $633 \mu\text{J}\cdot\text{cm}^{-2}$, the authors observe the progressive appearance of an early M-intermediate. This suggests that under $75 \mu\text{J}\cdot\text{cm}^{-2}$, bR undergoes its physiological photocycle, while at higher laser powers, laser-induced artefacts begin to accumulate.

As found in the discussion section of our manuscript, this is consistent with our own data. We kindly refer the referee to Supplementary table 2, where the crystallographic data recorded below $75 \mu\text{J}\cdot\text{cm}^{-2}$ ($0.04 \text{ GW}/\text{cm}^2$ and $343 \text{ GW}/\text{cm}^2$ power density) does not exhibit clear light-induced artefacts. However, data recorded with increasing laser power density show a decline in the quality of difference map signals, as light induced artefacts accumulate. It is likely that our data recorded at $525 \text{ GW}/\text{cm}^2$ and $1281 \text{ GW}/\text{cm}^2$ belongs to the group 2 described above, where the artifacts accumulate but remain below the threshold for visible detection in crystallographic data, manifesting instead as diminished map quality. Finally, the data recorded at $2493 \text{ GW}/\text{cm}^2$, likely belonging to group 3, exhibit visible artefact.

This suggests that while our results correlate with the spectroscopic findings presented by Engilberge *et al.*¹, the thresholds at which artifacts may become visible may vary between spectroscopic and crystallographic techniques. We believe that a femtosecond laser spectroscopic study on bR protein crystals would be a valuable addition to our data. However, this would require the development of a new dedicated experimental setup, which we hope is being pursued by spectroscopy-focused laboratories. Once available, we will be eager to use such infrastructure to improve our experiments further.

For additional discussion on the mentioned light scattering effects, please refer to our earlier comments.

The final main conclusion by the authors is that structurally, the protein is much less sensitive to changes from increased laser powers in crystals than spectroscopically in solution. Can the authors comment on the relevance of this assertion? Is this a crystalline effect? Shouldn't the comparison be with spectroscopy in crystals?

This effect is attributed to the experimental setup, as mentioned in the article. The dense packing of light sensitive molecules within the crystal reduces the photon density that penetrates through successive crystal layers, making the laser power a lot less precisely determined compared to a spectroscopy experiment. Laser power values used as cutoff for light-induced artifact creation are typically derived from in-solution spectroscopy assays, which can lead to biased interpretation since both techniques have different sensitivity.

We agree that the ideal comparison would be with spectroscopy on crystals; however, this is not a straightforward task, especially at the power densities used and in a femtosecond setup. This is beyond the scope of our study. Instead, we instead reference other published works on crystals and solution, as discussed earlier.

Is there actually changes that happen at longer timescales from multiphoton absorption not commented on in this work?

As mentioned in the article, we focused, here, on the K-intermediate, as this is the first ground state intermediate and well-studied by crystallography and spectroscopy. While it is possible that multiphoton absorption could structurally impact longer timescales, drawing such conclusions would require datasets at various laser powers and longer time points, which have not yet been collected. Since femtosecond lasers are not available at

synchrotron beamlines, the necessary experiments would go far beyond the scope of our article but would nevertheless be interesting.

However, according to Engilberge *et al.*¹, high laser power appears to induce the accumulation of an early M-intermediate, which may correspond to the L-like additional retinal conformation we observe at high laser power. This aligns with our findings on heat dissipation in the crystals, as increased temperature would accelerate reaction kinetics.

We added a sentence:

“The study shows that high laser fluence increases the accumulation of early M-intermediate¹, which aligns with our results, where the laser-induced temperature jump would affect photocycle kinetics.”

Minor suggestion:

It is unclear what the correlation between the nanosecond laser data and the femtosecond laser data should be in figure 3. Please clarify in the text for the non-expert.

The figure 3 does not show any ns laser data, and we believe the reviewer may have confused it with Figure 2. To clarify, we added indications in the main text.

“This series, collected at the same intermediate laser fluence, allows to track global laser effects by comparing relative atomic B-factors and changes in unit cell volume over time. This provides insights about the experimental impact of delivering the same photon dose with either a short (femtosecond) or long (nanosecond) laser pulse.”

Reviewer #4 (Remarks to the Author):

We think including early career researchers in review work, teaching them but also helping overloaded referees is an excellent idea. We would like to thank all reviewers for their time, insights and the interest in our work.

References:

- 1 Engilberge, S. *et al.* The TR-icOS setup at the ESRF: time-resolved microsecond UV-Vis absorption spectroscopy on protein crystals. *Acta Crystallogr D Struct Biol* **80**, 16-25 (2024). <https://doi.org:10.1107/S2059798323010483>
- 2 Fransson, T. *et al.* X-ray Emission Spectroscopy as an in Situ Diagnostic Tool for X-ray Crystallography of Metalloproteins Using an X-ray Free-Electron Laser. *Biochemistry* **57**, 4629-4637 (2018). <https://doi.org:10.1021/acs.biochem.8b00325>
- 3 Nango, E. *et al.* A three-dimensional movie of structural changes in bacteriorhodopsin. *Science* **354**, 1552-1557 (2016). <https://doi.org:10.1126/science.aah3497>
- 4 Claesson, E. *et al.* The primary structural photoresponse of phytochrome proteins captured by a femtosecond X-ray laser. *eLife* **9**, e53514 (2020). <https://doi.org:10.7554/eLife.53514>
- 5 Nass Kovacs, G. *et al.* Three-dimensional view of ultrafast dynamics in photoexcited bacteriorhodopsin. *Nat Commun* **10**, 3177 (2019). <https://doi.org:10.1038/s41467-019-10758-0>
- 6 Gruhl, T. *et al.* Ultrafast structural changes direct the first molecular events of vision. *Nature* **615**, 939-944 (2023). <https://doi.org:10.1038/s41586-023-05863-6>
- 7 Do, T. N., Menendez, D., Bizhga, D., Stojkovic, E. A. & Kennis, J. T. M. Two-photon Absorption and Photoionization of a Bacterial Phytochrome. *J Mol Biol*, 168357 (2023). <https://doi.org:10.1016/j.jmb.2023.168357>
- 8 Pandey, S. *et al.* Time-resolved serial femtosecond crystallography at the European XFEL. *Nat Methods* **17**, 73-78 (2020). <https://doi.org:10.1038/s41592-019-0628-z>
- 9 Efremov, R., Gordeliy, V. I., Heberle, J. & Buldt, G. Time-resolved microspectroscopy on a single crystal of bacteriorhodopsin reveals lattice-induced differences in the photocycle kinetics. *Biophys J* **91**, 1441-1451 (2006). <https://doi.org:10.1529/biophysj.106.083345>